# RoboEXP: Action-Conditioned Scene Graph via Interactive Exploration for Robotic Manipulation

**Hanxiao Jiang**[1,2]  **Binghao Huang**[1]  **Ruihai Wu**[4]  **Zhuoran Li**[5]
**Shubham Garg**[3]  **Hooshang Nayyeri**[3]  **Shenlong Wang**[2]  **Yunzhu Li**[1]
[1]Columbia University   [2]University of Illinois Urbana-Champaign   [3]Amazon
[4]Peking University   [5]National University of Singapore

**Abstract:** We introduce the novel task of interactive scene exploration, wherein robots autonomously explore environments and produce an action-conditioned scene graph (ACSG) that captures the structure of the underlying environment. The ACSG accounts for both low-level information (geometry and semantics) and high-level information (action-conditioned relationships between different entities) in the scene. To this end, we present the Robotic Exploration (RoboEXP) system, which incorporates the Large Multimodal Model (LMM) and an explicit memory design to enhance our system's capabilities. The robot reasons about what and how to explore an object, accumulating new information through the interaction process and incrementally constructing the ACSG. Leveraging the constructed ACSG, we illustrate the effectiveness and efficiency of our RoboEXP system in facilitating a wide range of real-world manipulation tasks involving rigid, articulated objects, nested objects, and deformable objects. Project Page: https://jianghanxiao.github.io/roboexp-web/

**Keywords:** Action-Conditioned Scene Graph, Foundation Models for Robotics, Scene Exploration, Robotic Manipulation

## 1  Introduction

Imagine a household robot designed to prepare breakfast. This robot must perform various tasks such as conducting inventory checks in cabinets, fetching food from the fridge, gathering utensils from drawers, and spotting leftovers under food covers. Key to its success is the ability to interact with and explore the environment, especially to find items that aren't immediately visible. Equipping it with such capabilities is crucial for the robot to effectively complete its everyday tasks. Robot exploration and active perception have long been challenging areas in robotics. Various techniques have been proposed, including information-theoretic approaches [1–7], frontier-based methods [8–10], imitation learning [11, 12] and reinforcement learning [13–17]. Nevertheless, previous research has primarily focused on exploring static environments by merely changing viewpoints in a navigation setting.

In this work, we investigate the interactive scene exploration task, where the goal is to efficiently identify all objects, including those that are directly observable and those that can only be discovered through interaction between the robot and the environment (Fig. 1). Towards this goal, we present a novel scene representation called action-conditioned 3D scene graph (ACSG). Unlike conventional 3D scene graphs that focus on encoding static relations, ACSG encodes both spatial relationships and logical associations indicative of action effects (e.g., opening a fridge will reveal an apple inside). We then show that our task can be formulated as a problem of ACSG construction and traversal.

To tackle the interactive scene exploration task, we propose a novel, real-world robotic exploration framework, the RoboEXP system. At the core of our system is a large foundational model-powered instantiation of action-conditioned 3D scene graph. Specifically, our framework consists of four modules: perception, memory, decision-making, and action (Fig. 3).

RoboEXP can handle diverse exploration tasks in a zero-shot manner, constructing complex action-conditioned 3D scene graph in various scenarios, including those involving obstructing objects and

8th Conference on Robot Learning (CoRL 2024), Munich, Germany.

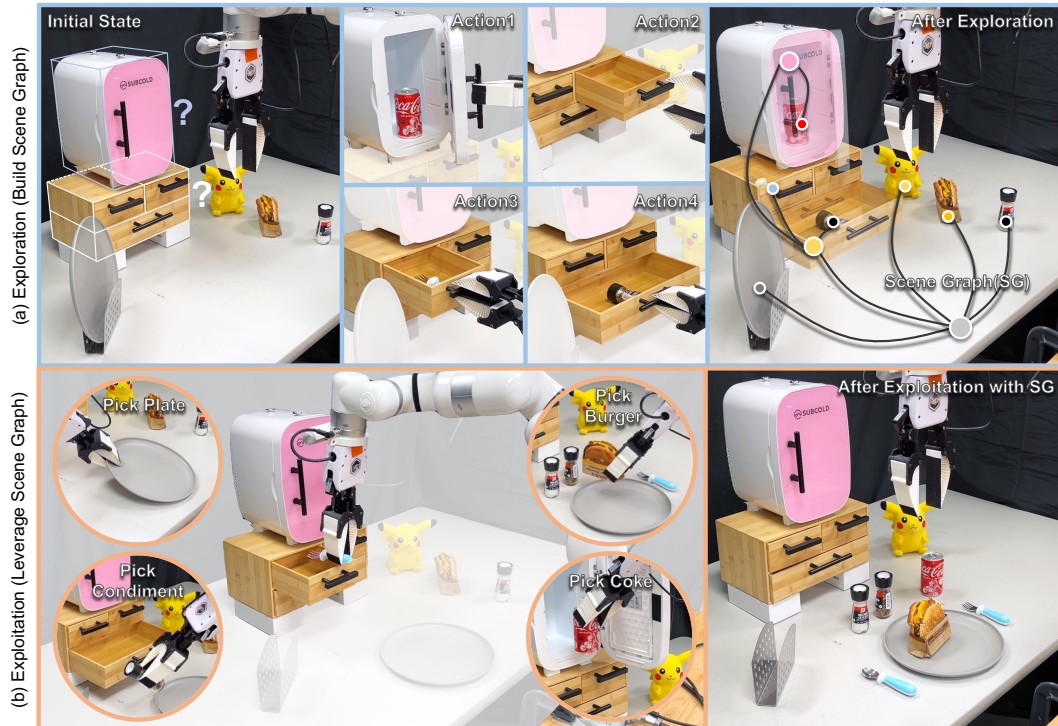

Figure 1: **Interactive Exploration to Construct an Action-Conditioned Scene Graph (ACSG) for Robotic Manipulation.** (a) **Exploration:** The robot autonomously explores by interacting with the environment to generate a comprehensive ACSG. This graph is used to catalog the locations and relationships of items. (b) **Exploitation:** Utilizing the constructed scene graph, the robot completes downstream tasks by efficiently organizing the necessary items according to the desired spatial and relational constraints.

requiring multi-step reasoning (Fig. 2). We evaluate our system across various settings , demonstrating its adaptability and robustness. The system also effectively manages different human interventions. Moreover, we show that our reconstructed action-conditioned 3D scene graph demonstrates strong capacity in performing multiple complex downstream tasks. Action-conditioned 3D scene graph advances LLM/LMM-guided robotic manipulation and decision-making research [18, 19], extending their operation domain from environments with known or observable objects to complicated environments with unknown or unobserved ones. To our knowledge, this is the first of its kind.

## 2 Related Works

**Scene Graphs** [20, 21] represent objects and their relations [22–24] in a scene via a graph structure. Previous studies generate scene graphs from images [21, 25, 26] or 3D scenes [27], and further with the assistance of large language models (LLMs) [28, 29]. While previous works model scene graphs in static 2D or 3D scenes, we generate action-conditioned scene graphs that integrate actions as core elements, depicting interactive relationships between objects and actions. Our work is also related to **Neuro-Symbolic Representations** that integrate neural networks with symbolic reasoning. Prior works explored understanding scenes and describing robotic skills in symbolic texts to interpret demonstrations [30, 31], ground abstract actions for robotic primitives [32] and generate action plans [33–38]. Our proposed framework also constructs symbolic representations of the environment, but in the form of action-conditioned scene graphs for robotic manipulation.

**Robotic Exploration** aims to autonomously navigate [9, 11, 39–45], interact with [12, 46, 46–50], and gather information [51–53] from environments it has never encountered before. The primary guiding principle behind robotic exploration is to reduce the uncertainty of the environment [5–7, 39, 54, 55], making uncertainty quantification key for robotic exploration tasks. Curiosity-driven exploration has recently emerged as a promising approach, showing effective results in various contexts [16, 56–58]. Recently, exploration has also been studied in the context of manipulation [59–66], aiming to better understand the scene by changing the state of the environment. Our work introduces a new active exploration strategy for manipulation, uniquely defining a novel scene graph-guided objective to guide the exploration process. Our work is also related to **Active Perception**,

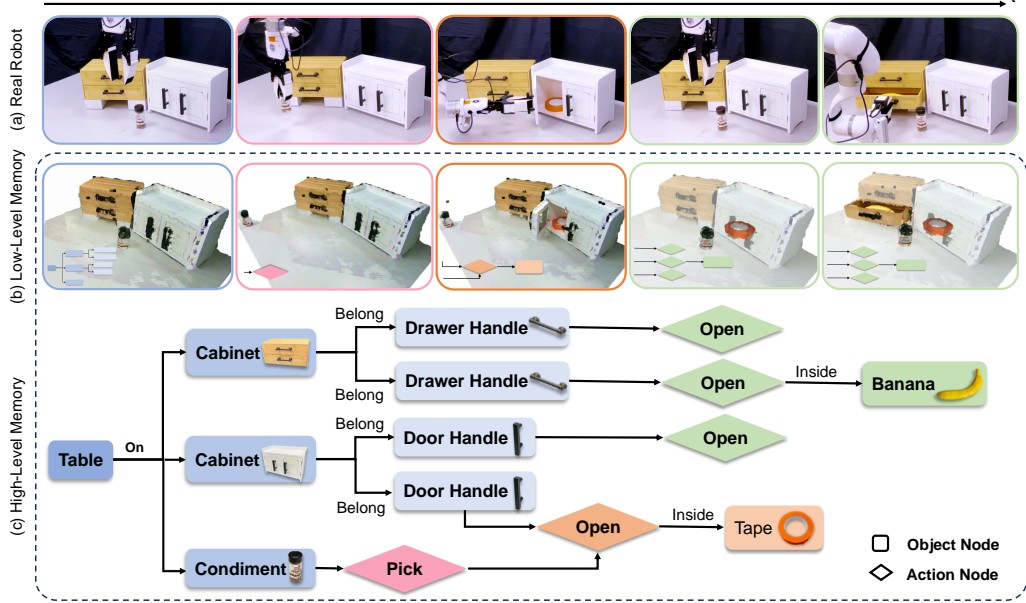

Figure 2: **Action-Conditioned 3D Scene Graph from Interactive Scene Exploration.** We depict a scenario wherein a robot arm explores a tabletop scene containing two cabinets and a condiment obstructing the left door. (a) The robot arm actively interacts with the scene, completing the interactive scene exploration process. (b) We showcase the corresponding low-level memory in our ACSG. The small graph on the bottom-left of each visualization represents a segment of the final scene graph. (c) We present the high-level memory of our ACSG. The graph reveals that picking up the condiment serves as a precondition for opening the door, and opening the bottom drawer allows the observation of the concealed banana.

which actively selects actions for an agent to help it perceive and understand the environment [1, 67]. Unlike passive perception, actions offer more flexibility, such as control over viewpoints [2–4, 68], sensor configurations [15, 69], or adjustments to environmental configurations [70]. It can also reveal certain scene properties that cannot be perceived in a passive manner, such as dynamic parameters [16, 71] or articulation [62, 72, 73]. Our work falls into the category of actively exploring the environment to reveal what's inside or underneath objects. Differing from most previous active perception efforts, which are driven by handcrafted rules [74], information gain [75, 76], or reinforcement learning [16, 77], our approach is guided by grounding the commonsense knowledge encoded in a LLM/LMM into an explicit scene graph representation.

**Language Models for Robotics.** Large language models (LLMs) [78–80] and large multimodality models (LMMs) [81, 82] are bringing overwhelming influence into the robotics field, for their strong capacity in common-sense knowledge and task-level planning. Previous studies have harnessed the common-sense knowledge of such large models to generate action candidates [83–85] and action sequences for task planning [80, 86–88], and generate code for robotic control and manipulation [18, 89, 90]. More recently, VILA [19] utilized GPT-4V [81, 82] for vision-language planning. In our RoboEXP system, we leverage GPT-4V for decision-making in two crucial roles to select and verify actions. Moreover, instead of memorizing everything using large models in a brute force way, our system employs explicit memory to enhance the decision-making process.

## 3 Action-Conditioned 3D Scene Graph

An action-conditioned 3D scene graph (ACSG) is an actionable, spatial-topological representation that models objects and their interactive and spatial relations in a scene. Formally, ACSG is a directed acyclic graph $\mathbf{G} = (\mathbf{V}, \mathbf{E})$ where each node represents either an object or an action, and edges $\mathbf{E}$ represent their interaction relations. The object node $\mathbf{o}_i = (\mathbf{s}_i, \mathbf{p}_i) \in \mathbf{V}$ encodes the semantics $\mathbf{s}_i$ and geometry $\mathbf{p}_i$ of each object, whereas the action node $\mathbf{a}_k = (a_k, \mathbf{T}_k) \in \mathbf{V}$ encodes high-level action type $a_k$ and low-level primitives $\mathbf{T}_k$ to perform the actions. Between the nodes are edges encoding their relations, which we categorize into four types: 1) between objects $\mathbf{e}_{\mathbf{o} \to \mathbf{o}}$ (e.g., the *door handle belongs* to the *fridge*), 2) from objects to actions $\mathbf{e}_{\mathbf{o} \to \mathbf{a}}$ (e.g., *toy* can be *picked up*), 3) from action to objects $\mathbf{e}_{\mathbf{a} \to \mathbf{o}}$ (e.g., a *banana* can be reached if we *open* the cabinet), or 4) from one

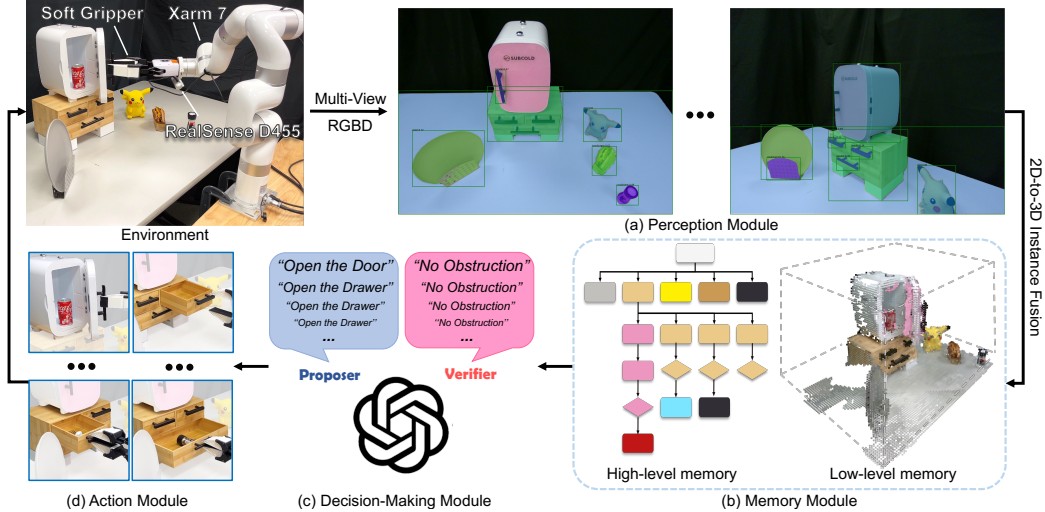

Figure 3: **Overview of Our RoboEXP System.** We present a comprehensive overview of our RoboEXP system, comprised of four modules: (a) perception, (b) memory, (c) decision-making and (d) action module.

action to another $\mathbf{e_{a \to a}}$ (e.g., the cabinet can be *opened* only if we *move away* the *condiment*). Our action-conditioned 3D scene graph greatly enhances existing 3D scene graphs, as it explicitly models the action-conditioned relations between objects. Fig. 2 depicts a complete action-conditioned 3D scene graph of a tabletop scene.

One advantage of our ACSG lies in its simplicity for retrieving and taking actions on an object. Regardless of how complicated the scene is, given our scene graph and a target object, an agent merely needs to sequentially execute all the actions on the paths from the root to the object node in a topological order to retrieve the object.

**Interactive Exploration.** Constructing a complete ACSG of a real-world scene is challenging due to partial observability. To solve this task, we formulate the scene graph construction as an active perception and exploration problem using POMDP-inspired notations. Formally, at each time $t$, based on our past graph estimation $\mathbf{G}^{t-1}$, and past sensor observations $\mathbf{O}^{t-1}$, our agent takes an action $\mathbf{A}^t$, which causes the environment to transition to a new state, and the agent receives a new observation $\mathbf{O}^t$, which is used to update its current inferred graph $\mathbf{G}^t$. This update might include adding new nodes to the graph or updating the state of an existing node. We will then continue with exploration and keep updating the set of remaining unexplored nodes $\mathbf{U} \subset \mathbf{V}$ (See the Appendix for the full algorithm). The goal of the exploration is simple: discover and explore all the nodes of the scene graph in as little time as possible. We thus formulate a reward function with three terms: $\mathbf{R}^t = \mathbf{R}^t_{\text{graph}} + \mathbf{R}^t_{\text{explore}} + \mathbf{R}^t_{\text{time}}$, where $\mathbf{R}^t_{\text{graph}} = |\mathbf{V}^t| - |\mathbf{V}^{t-1}|$ promotes our agent to discover as many nodes as possible to the graph, $\mathbf{R}^t_{\text{explore}} = \max(0, |\mathbf{U}^{t-1}| - |\mathbf{U}^t|)$ gives a positive reward to actions that reduce the unexplored node set, which prioritizes the agent to explore previously unexplored nodes (note that an action can lead to the discovery of new unexplored nodes being added to the graph), and immediate reward $\mathbf{R}^t_{\text{time}} = -\lambda, 0 < \lambda < 1$ is a negative time reward that enhances efficiency and terminate the exploration when there is no more node to explore.

Intuitively, to maximize this reward at each discrete timestamp, we should prioritize exploring the unexplored nodes in the current scene graph that are likely to lead to the discovery of new nodes (e.g., opening a cabinet that has not been opened, or lifting a piece of clothing that might cover a small object). The key challenge lies in how we can perceive the objects in the scene, infer possible actions and their relations from the sensory data, and take actions with the current scene graph. In the next section, we will comprehensively describe our system implementation to achieve this goal.

## 4 Interactive Robotic Exploration (RoboEXP) System

In this section, we outline the structure of our RoboEXP system, including perception, memory, decision-making, and action modules (See the Appendix for full details of our implementation).

**Perception Module.** Given multiple RGBD observations from different viewpoints, the objective of the perception module (Fig. 3a) is to detect and segment objects while extracting their semantic

features. To enhance generality, we opt for the open-vocabulary detector GroundingDINO [91] and the Segment Anything in High Quality (SAM-HQ) [92], with a predefined list of tags. For the extraction of semantic features used in subsequent instance merging, we employ per-instance CLIP [93], following a similar strategy to the one proposed by Jatavallabhula et al. [94].

**Memory Module.** The memory module (Fig. 3b) handles merging across different viewpoints and time steps. To merge across different viewpoints, we project 2D information to 3D and leverage the instance merging strategy similar to Lu et al. [95] to attain consistent 3D information. Addressing memory updates across time steps presents a challenge due to dynamic changes in the environment. For instance, a closed door in the previous time step may be opened by our robot in the current time step. To accurately reflect such changes, our algorithm evaluates whether elements within our memory have become outdated, primarily through depth tests based on the most recent observations. This process ensures that the low-level memory accurately represents the environment's current state, effectively managing scenarios where objects may change positions or states across different time steps. For the high-level graph of our ACSG, the memory module analyzes the relationships between objects and the logical associations between actions and objects. Depending on changes in low-level memory and relationships, the memory module is tasked with updating the graph. This involves adding, deleting, or modifying nodes and edges within our graph.

**Decision-Making Module.** The primary goal of the decision module (Fig. 3c) is to identify the appropriate object and corresponding skill to enhance the effectiveness and efficiency of interactive scene exploration. In the context of our task, distinct objects may necessitate distinct exploration strategies. While humans can easily discern the most suitable skill to apply (e.g., picking up the top Matryoshka doll to inspect its contents), achieving such decisions through heuristic-based methods is challenging. The LMM brings commonsense knowledge to our decision-making process and serves in two pivotal roles. Firstly, it functions as an action proposer. Given the current digital environment from the memory module, GPT-4V is tasked with selecting the appropriate skill for unexplored objects in our system. Secondly, the LMM also serves as the action verifier. For the proposer role, it analyzes the object-centric attributes but ignore surrounding information when choosing the proper skill. To address this, we use another LMM program to verify the feasibility of the action and identify any objects in the scene that may impede the action based on information from our ACSG.

**Action Module.** In the action module (Fig. 3d), our primary focus is on autonomously constructing the ACSG through effective and efficient interaction. We design a set of action primitives that operate on the low-level geometric information embedded within our ACSG. These primitives encompass seven categories: "open the door", "open the drawer", "close the door", "close the drawer", "pick object to idle space", "pick back object", "move wrist camera to position". Strategic utilization of these skills plays a pivotal role in accomplishing intricate tasks seamlessly within our system.

## 5 Experiments

In this section, we assess the performance of our system across a variety of tabletop scenarios. Our primary objective is to address two key questions through experiments: 1) How does our proposed system handle diverse exploration scenarios well and build the complete ACSG? 2) How useful is our ACSG in the downstream object retrieval and manipulation tasks?

### 5.1 Interactive Exploration and Scene Graph Building

**Robot Setups.** All our experiments are conducted in a real-world setting. In the tabletop scenarios, we mount one RealSense-D455 camera on the wrist of the robot arm to collect RGBD observations, with the execution of actions performed by the UFACTORY xArm 7. The end effector for our robot arm is the soft gripper (see Fig. 3). In the mobile setting, we choose the Hello Robot Stretch2 with the official upgraded kits. See the Appendix for more details on the full setup.

**Experiment Settings.** To assess our system's efficacy, we designed five types of experiments, each with 10 different settings varying in object number, type, and layout. We validate the performance of our system in constructing ACSG through quantitative analysis and qualitative demonstrations.

Table 1: **Quantitative Results on Different Tasks.** We compare the performance of both the GPT-4V baseline and our system across various tasks. Our system consistently outperforms the baseline across all metrics.

| Task (10 variance for each) | Drawer-Only | | Door-Only | | Drawer-Door | | Recursive | | Occlusion | |
|---|---|---|---|---|---|---|---|---|---|---|
| Metric | GPT-4V | Ours | GPT-4V | Ours | GPT-4V | Ours | GPT-4V | Ours | GPT-4V | Ours |
| Success % ↑ | 20±13.3 | **90**±10.0 | 30±15.2 | **90**±10.0 | 10±10.0 | **70**±15.3 | 0±0.0 | **70**±15.3 | 0±0.0 | **50**±16.7 |
| Object Recovery % ↑ | 83±11.0 | **97**±3.3 | 50±16.7 | **100**±0.0 | 62±10.7 | **91**±4.7 | 20±13.3 | **80**±11.7 | 17±11.4 | **67**±14.9 |
| State Recovery % ↑ | 60±16.3 | **100**±0.0 | 80±13.3 | **100**±0.0 | 70±15.3 | **100**±0.0 | 70±15.3 | **100**±0.0 | 10±10.0 | **70**±15.3 |
| Unexplored Space % ↓ | 15±7.6 | **0**±0.0 | 40±14.5 | **0**±0.0 | 25±6.5 | **0**±0.0 | 63±15.3 | **15**±8.9 | 85±7.6 | **30**±15.3 |
| Graph Edit Dist. ↓ | 2.8±1.04 | **0.2**±0.20 | 4.4±1.42 | **0.1**±0.10 | 5.6±1.46 | **0.5**±0.27 | 8.8±2.06 | **2.1**±1.49 | 7.3±0.97 | **2.5**±1.15 |

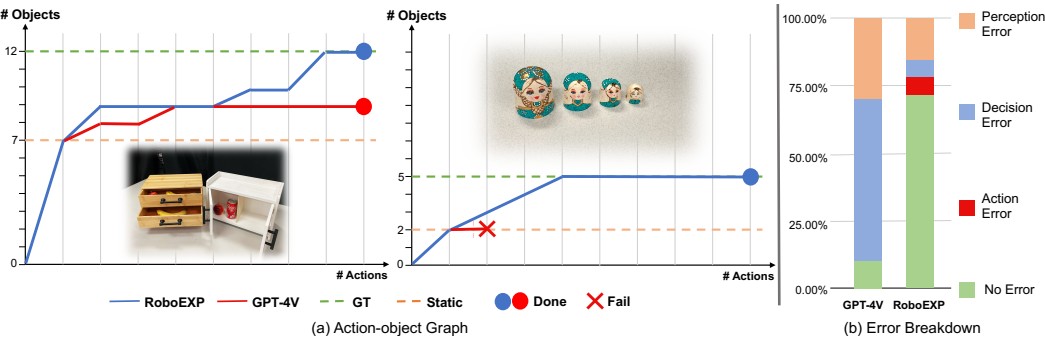

Figure 4: **Visualization of Quantitative Results.** (a) The action-object graph captures the change in the number of discovered objects relative to the number of actions taken. Our RoboEXP efficiently discovers all objects. (b) The error breakdown of all our quantitative experiments includes 5 task settings with 10 variations each. We categorize errors into perception, decision, action, and no-error cases. For the GPT-4V baseline, we manually assist in action execution, eliminating action errors. This serves as an upper bound for baseline performance. However, even with this enhancement, our RoboEXP consistently shows superior performance.

We compare our system with a strong baseline by augmenting GPT-4V with ground truth actions. This baseline operates in a closed-loop fashion, receiving RGB observations from different viewpoints. At each turn, it generates the current scene graph, encompassing hidden objects, and suggests the next action to be taken (refer to the complete prompts in the Appendix). To ensure the baseline is robust, we utilize manual actions as ground truth references for the proposed actions. In contrast, in the exploration experiments described below, all actions from our system are automatically executed by our action module on the physical robot. It is also crucial to note that the output ACSG from our system faithfully aligns with the task requirements. Conversely, for the baseline, we manually construct ACSG based on its actions and the new observations it uncovers. Due to the unstructured nature of the raw scene graph from the baseline, we carefully refine it according to the observable objects, providing an upper-bound performance for comparison during evaluation. (See comparisons with other heuristic-based baselines in Appendix)

We design five key metrics to measure the performance of the interactive exploration task. 1) **Success** evaluates the success percentage across 10 variants for each task. We define success for each experiment as 1 when the final outputted ACSG exactly matches the GT version, and 0 otherwise; 2) **Object Recovery** quantifies the percentage of hidden objects successfully identified; 3) **State Recovery** indicates whether the final state resembles the original state before exploration; 4) **Unexplored Space** evaluates the percentage of successfully explored need-to-explore space to reduce the robot's uncertainty about the scene; 5) **Graph Edit Distance (GED)** measures the disparity between the outputted graph and the GT graph.

**Comparison.** The quantitative findings presented in Tab. 2 underscore the superior performance of our system compared to the baseline method. Our approach showcases a notable enhancement across all metrics, outperforming the baseline by a considerable margin. The collective assessment of success rate, object recovery, and unexplored space metrics unequivocally validates the efficacy of our system in exploring unfamiliar scenes through interactive processes. It is essential to highlight that, in the case of object recovery, the baseline method may occasionally choose to randomly open certain drawers or doors, revealing objects that have already been discovered. This randomness contributes to a seemingly high object recovery rate but may not necessarily correlate with overall success. The unexplored space metric shows that our system is much more stable in exploring all need-to-explore spaces. Moreover, both the success rate and graph edit distance underscore the close alignment of our system with human actions, highlighting the efficiency of our approach across diverse scenarios. The state recovery metric assesses whether the final state post-exploration resembles the initial state. Our

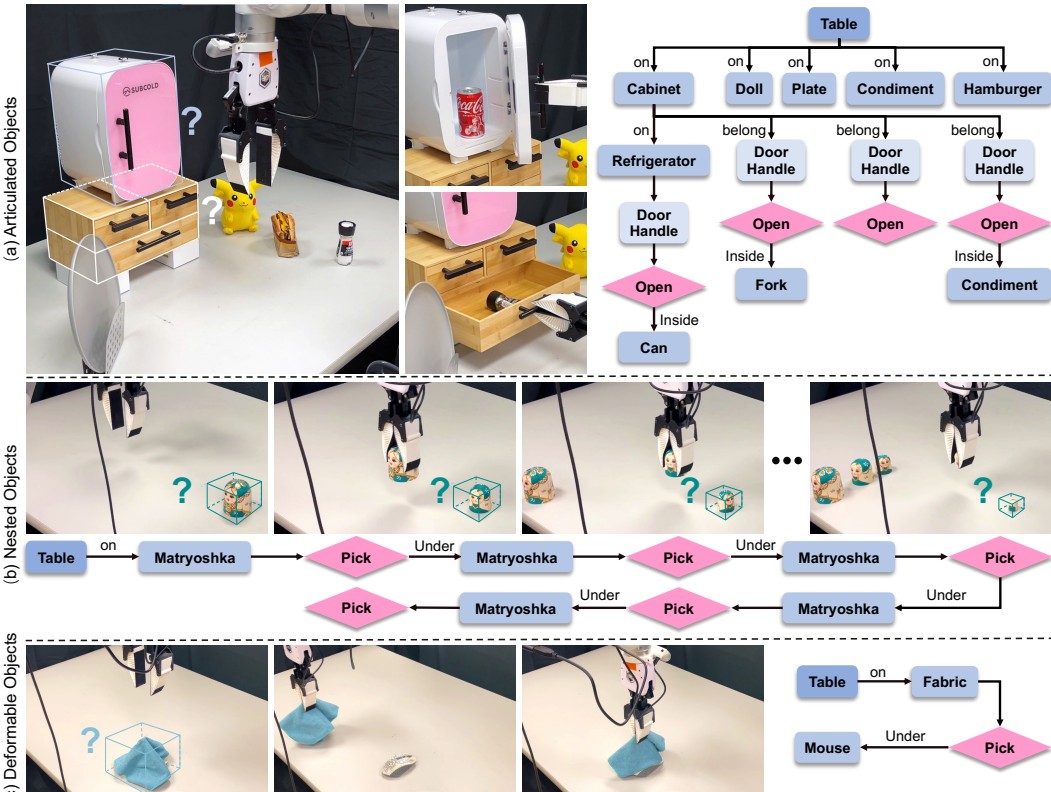

Figure 5: **Qualitative Results on Different Scenarios.** We visualize the interactive exploration process and the corresponding constructed ACSG. (a) This scenario involves a tabletop environment with two articulated objects, accompanied by additional items either on the table or concealed in storage space. The constructed scene graph demonstrates the success of our system in identifying all objects within the environment through a series of physical interactions. (b) This scenario includes nested objects, five Matryoshka dolls, with only the top one being directly observable. Our system autonomously decides to explore the contents through a recursive reasoning process, showcasing its ability to construct deep ACSG. (c) This scenario involves a fabric covering a mouse, showcasing exploration scenarios that involve a deformable object. Our system interacts with the fabric and successfully uncovers what lies beneath it.

system consistently shows effective state recovery; however, the baseline may trick this metric by opting not to take any action, resulting in an artificially high score in this aspect.

Fig. 4a provides additional insights, illustrating that as the number of actions increases, so does the number of objects. Specifically, we present the ground truth object number alongside the directly-observable object number that can be represented by the traditional 3D scene graph. These results underscore our system's ability to achieve robust and efficient exploration throughout the exploration process. Our system excels in efficiently discovering all concealed objects, whereas the baseline fails either due to a lack of early-stage actions or an inability to explore all need-to-explore spaces even upon completion. The analysis of errors (Fig. 4b) in both our system and the baseline reveals the specific failure cases encountered by the baselines. In contrast, our system demonstrates enhanced robustness in both perception and decision-making.

Fig. 5 further illustrates various exploration scenarios along with their corresponding ACSG. These scenarios encompass ACSG with varying width or depth, highlighting our system's adaptive capability across diverse objects such as rigid, articulated objects, nested objects, and deformable objects. In addition, the scenario in Fig. 2 shows that our system is able to deal with the scenario with obstruction.

## 5.2 ACSG for Object Retrieval and Manipulation Tasks

The scenarios depicted in Fig. 1 exemplify the efficacy of our generated output (ACSG) in manipulation tasks. Consider the table-rearranging scenario: without our ACSG, the robot struggles to swiftly prepare the table due to the lack of precise prior knowledge about the location of objects (e.g., the fork stored in the top-left drawer of the wooden cabinet). Beyond comprehensive layout guidance, our ACSG also addresses a crucial question regarding task feasibility for the robot. For

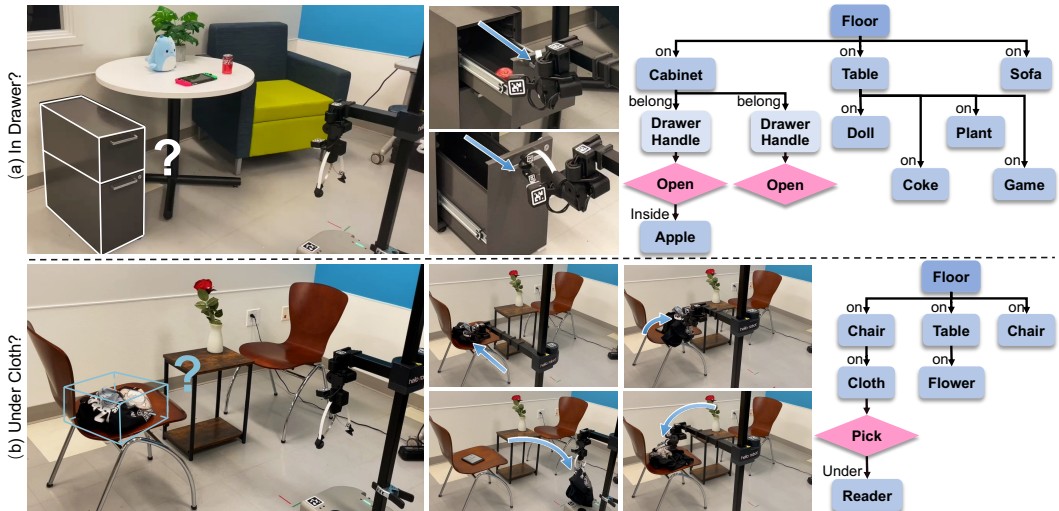

Figure 6: **Experiments on Mobile Robot Scenarios.** We visualize the interactive exploration process and the corresponding constructed ACSG in the mobile robot settings. (a) This scenario includes a table, a cabinet, a sofa, multiple small items, and an apple hidden in the top drawer. (b) This scenario involves two chairs, one table, and a reader hidden by the cloth above.

instance, if the task requires a spoon but there is no spoon in the scene, the robot recognizes the missing object and asks for human help. In addition to enhancing downstream manipulation tasks, our ACSG possesses the capability to autonomously adapt to environmental changes. In the human intervention setting, our system seamlessly explores newly added components, such as a cabinet, ensuring continuous adaptability. Check our Appendix and supplemental video for more details.

### 5.3 Extension to Mobile Robot

We also demonstrate the effectiveness of our pipeline using a mobile robot in household scenarios. To adapt to Stretch2, we merely modified our action module to follow the new kinematic structure, while keeping other modules nearly the same. Fig. 6 shows two scenes with hidden objects in drawers and under cloth. Our system is capable of exploring the scene, utilizing manipulation and mobility, to construct the full ACSG and recover the scene. By leveraging the constructed ACSG, we can easily locate the hidden apple and e-reader.

## 6 Limitations and Future Work

Although our system has proven effective, there is room for improvement. The breakdown of the failure rate in Fig. 4.b suggests that failures primarily arise from detection and segmentation errors within the perception module. To address this issue, we envision two future directions: 1) enhancing the capabilities of visual foundation models for open-world semantic understanding, and 2) utilizing temporal cues and semantic fusion techniques to improve perception robustness through continuous observations. Furthermore, our system would benefit from enhanced LMM capacities and the integration of more sophisticated and generalizable skill modules. Such improvements would enhance both the decision-making and the action execution, thereby further reducing failure cases.

## 7 Conclusion

We introduced RoboEXP, a foundation-model-driven robotic exploration framework capable of effectively identifying all objects in a complex scene, both directly observable and those revealed through interaction. Central to our system is action-conditioned 3D scene graph, an advanced 3D scene graph that goes beyond traditional models by explicitly modeling interactive relations between objects. Experiments have shown RoboEXP's superior performance in interactive scene exploration across various challenging scenarios, significantly outperforming a strong GPT4V-based baseline. Notably, the reconstructed action-conditioned 3D scene graph is crucial for guiding complex downstream manipulation tasks, like setting up the table in a mock environment with fridges, cabinets, and drawer sets. Our system and its action-conditioned scene graph lay the groundwork for practical robotic deployment in complex settings, especially in environments like households and offices, facilitating their integration into everyday human activities.

**Acknowledgments**

Yunzhu Li is partly supported by the Amazon AICE Award. Shenlong Wang is supported by the Amazon AICE Award, IBM IIDAI Grant, Nvidia Hardware Grants, the Insper-Illinois Innovative Grant, the NCSA Faculty Fellow, NSF Award #2331878 and #2312102, and gifts from Intel and Meta. We thank Dorie Wang for generously sharing her toys for our research, Kaifeng Zhang and Yixuan Wang for their kind discussions. This work does not relate to the positions of Shubham Garg and Hooshang Nayyeri at Amazon. The views and conclusions contained herein are those of the authors and should not be interpreted as necessarily representing the official policies, either expressed or implied, of the sponsors.

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

Our contributions are as follows: i) we propose action-conditioned 3D scene graph and introduce the interactive scene exploration task to address the challenging interaction aspect of exploration; ii) we develop the RoboEXP system, capable of exploring complicated environments with unseen objects in a wide range of settings; iii) through extensive experiments, we demonstrate our system's ability to construct complex and complete action-conditioned 3D scene graph, demonstrating significant potential for various manipulation tasks. Our experiments involve rigid and articulated objects, nested objects like Matryoshka dolls, and deformable objects like cloth, showcasing the system's generalization ability across objects, scene configurations, and downstream tasks.

## A   Additional Details of Interactive Exploration

Due to space constraints, we did not include the pseudocode of the algorithm proposed in the main paper, but include more details here for clarity. We formulate the interactive scene exploration task into an active perception and exploration problem to construct the action-conditioned 3D scene graph (ACSG).

---

**Algorithm 1** Interactive Exploration

---

1: **input: $\mathbf{O}^0, \mathbf{G}^0 = (\mathbf{V}^0, \mathbf{E}^0), \mathbf{U}^0 \leftarrow \mathbf{V}^0$**
2: **while** $|\mathbf{U}^{t-1}| \neq 0$ **do**
3:     **if** choose object $\mathbf{o}_i \in \mathbf{U}^{t-1}$ **then**              % explore object
4:         add spatial relations (Algorithm 2)                    % memory
5:         obtain action $\mathbf{a}$ to explore $\mathbf{o}_i$              % decision-making
6:         **if** action $\mathbf{a} \notin \mathbf{V}^{t-1}$ **then**
7:             $\mathbf{V}^t, \mathbf{U}^t = \mathbf{V}^{t-1} \cup \{\mathbf{a}\}, \mathbf{U}^{t-1} \cup \{\mathbf{a}\}$              % add node
8:             $\mathbf{E}^t = \mathbf{E}^{t-1} \cup \{\mathbf{e}_{\mathbf{o}_i \to \mathbf{a}}\}$              % add edge
9:             $\mathbf{U}^t = \mathbf{U}^t \setminus \mathbf{o}_i$              % mark as explored
10:         **end if**
11:     **else** choose action $\mathbf{a}_k \in \mathbf{U}^{t-1}$
12:         **if** no obstruction **then**              % decision-making
13:             take action $\mathbf{a}_k$              % action
14:             obtain new observation $\mathbf{O}^t$              % perception
15:             **if** found new objects $\mathcal{O} \not\subset \mathbf{V}^{t-1}$ **then**
16:                 $\mathbf{V}^t, \mathbf{U}^t = \mathbf{V}^t \cup \{\mathcal{O}\}, \mathbf{U}^{t-1} \cup \{\mathcal{O}\}$              % add nodes
17:                 $\mathbf{E}^t = \mathbf{E}^t \cup \{\mathbf{e}_{\mathbf{a}_k \to \mathcal{O}}\}$              % add edges
18:                 $\mathbf{U}^t = \mathbf{U}^t \setminus \mathbf{a}_k$              % mark as explored
19:             **end if**
20:         **else**
21:             add action preconditions (Algorithm 3)              % memory
22:         **end if**
23:     **end if**
24: **end while**
25: **output: $\mathbf{G}^t$**              % final scene graph

---

In the above algorithm, we have demonstrated how to construct the edges from objects to actions $\mathbf{e}_{\mathbf{o} \to \mathbf{a}}$ and from actions to objects $\mathbf{e}_{\mathbf{o} \to \mathbf{a}}$; however, there is a lack of description for the other two types of edges.

**Add Spatial Relations.** The logic involves analyzing the spatial relationships among objects using spatial heuristics and incorporating the resulting spatial relation edges between objects $\mathbf{e}_{\mathbf{o} \to \mathbf{o}}$ (Algorithm 2).

**Add Action Preconditions.** The approach is to assess the feasibility of implementing the actions. We utilize the decision-making module to verify whether there are any prerequisite actions that need to be completed beforehand, and then adjust the plan accordingly (Algorithm 3).

**Algorithm 2** Add Spatial Relations

---

1: **input:** $\mathbf{G}^{t-1} = (\mathbf{V}^{t-1}, \mathbf{E}^{t-1})$
2: $\mathbf{E}^t = \mathbf{E}^{t-1}$
3: **for** $\mathbf{o} \in \mathbf{V}^{t-1}$ **do**      % check relations
4:    **if** relation from $\mathbf{o}$ to $\mathbf{o}_i$ **then**      % memory
5:       $\mathbf{E}^t = \mathbf{E}^t \cup \{\mathbf{e}_{\mathbf{o} \to \mathbf{o}_i}\}$      % add edge
6:    **end if**
7:    **if** relation from $\mathbf{o}_i$ to $\mathbf{o}$ **then**
8:       $\mathbf{E}^t = \mathbf{E}^t \cup \{\mathbf{e}_{\mathbf{o}_i \to \mathbf{o}}\}$      % add edge
9:    **end if**
10: **end for**
11: **output:** $\mathbf{G}^t$      % new scene graph

---

**Algorithm 3** Add Action Preconditions

---

1: **input:** $\mathbf{G}^{t-1} = (\mathbf{V}^{t-1}, \mathbf{E}^{t-1}), \mathbf{U}^{t-1}$
2: **if** object $\mathbf{o}$ obstruct **then**      % decision-making
3:    choose action $\mathbf{a}$
4:    $\mathbf{V}^t = \mathbf{V}^{t-1} \cup \{\mathbf{a}\}, \mathbf{U}^{t-1} \cup \{\mathbf{a}\}$      % add node
5:    $\mathbf{E}^t = \mathbf{E}^{t-1} \cup \{\mathbf{e}_{\mathbf{o} \to \mathbf{a}}\}$      % add edge
6:    $\mathbf{E}^t = \mathbf{E}^{t-1} \cup \{\mathbf{e}_{\mathbf{a} \to \mathbf{a}_k}\}$      % add edge
7: **end if**
8: **output:** $\mathbf{G}^t, \mathbf{U}^t$      % new scene graph & plan

---

# B   Additional Details of RoboEXP System

We provide additional details about our system and each module in it. We then discuss our system's design for the interactive scene exploration task, focusing on its application in closed-loop exploration processes that may require multi-step or recursive reasoning and handle potential interventions. Additionally, we explain the usage of our proposed ACSG.

## B.1   Details of Modules in RoboEXP System

To tackle the interactive exploration task, we present our RoboEXP system, designed to autonomously explore unknown environments by observing and interacting with them. The system comprises four key components: perception, memory, decision-making, and action modules. This closed-loop system ensures the thoroughness of our task in interactive scene exploration.

### B.1.1   Perception Module

Raw RGBD images are captured through the wrist camera in different viewpoints and processed by the perception modules to extract scene semantics, including object labels, 2D bounding boxes, segmentations, and semantic features. As mentioned in the main paper, to obtain per-instance CLIP features, we implement a strategy similar to the one proposed by Jatavallabhula et al. [94]. Specifically, we extend the local-global image feature merging approach by incorporating additional label text features to augment the semantic CLIP feature for each instance. Furthermore, we exclusively focus on instance-level features, disregarding pixel-level features, thereby accelerating the entire semantic feature extraction process.

### B.1.2   Memory Module

The memory module is designed to construct our ACSG of the environment by assimilating observations over time. For the low-level memory, to ensure stable instance merging from 2D to 3D, we employ a similar instance merging strategy as presented in Lu et al. [95], consolidating observations from diverse RGBD sources across various viewpoints and time steps. In contrast to the original

algorithm, which considers only 3D IoU and semantic feature similarity we additionally incorporate label similarity and instance confidence. To enhance algorithm efficiency, we represent low-level memory using a voxel-based representation with filtering designs, which allows for more efficient computation and cleaner memory updates. Meanwhile, given the crowded nature of objects in our tabletop setting, we have implemented voxel-based filtering designs to obtain a cleaner and more complete representation of the objects for storage in our memory.

**(a) Prompts of Proposer**

**System:** You are an assistant tasked with aiding in the construction of a complete scene graph for a tabletop environment. The objective is to identify all objects hidden from the current observation in the tabletop setting. Your role involves selecting appropriate actions or opting not to take any action based on commonsense knowledge in response to queries with current observations. Your responses will guide a robot in efficiently exploring the environment. Approach each step thoughtfully, and analyze the fundamental problem deeply, considering the potential vagueness or inaccuracy in the queries. Adhere to the provided formats in your instructions.

**User:** Analyze and provide your final answer for each new query object/part category, considering the given surrounding objects and observations in the tabletop scene from different viewpoints. The query object/part will be enclosed in a green bounding box, though it may not always be fully accurate. Format your responses as follows: "[Analysis]: <your reasoning process>; \n\n [Final Answer]: <skill>". Be comprehensive and avoid repeating my question. Choose from three skills: 1. Open the doors or drawers. 2. Pick up / Open the top object. 3. No action. The primary goal is to select an action that has the potential to reveal hidden objects. The secondary goal is to act efficiently, performing only necessary actions to uncover hidden objects. For example, if an object contains doors or drawers and can potentially store something inside, opt for the first skill "Open the doors or drawers". If an object has no bottom side and can potentially cover something beneath it, choose the second skill " Pick up / Open the top object"; otherwise, select the third skill "No action" to ensure efficiency.

**Assistant:** Got it. I will output the reasoning process step-by-step, explain why I choose the skill but not others and follow the output format.

**User:** [Query Object] + [Query Images]

**Assistant:** [Reply from GPT-4V]

**(b) Prompts of Verifier**

**System:** You are an assistant tasked with evaluating the feasibility of actions within a tabletop environment. Your role is to select suitable objects that could obstruct open actions based on queries and current observations. Provide guidance for a robot's planning process. Approach each step thoughtfully, analyzing the underlying problem thoroughly while considering potential vagueness or inaccuracy in the queries. Follow the provided formats in your instructions.

**User:** Provide an analysis and your final answer each time I present a new query object/part category, the list of surrounding objects you need to consider and observations of the corresponding in the tabletop scene from different viewpoints. The query object/part is enclosed in a green bounding box, which may not always be fully accurate. Present your reasoning process and final answer in the format "[Analysis]: <your reasoning process>; \n\n [Final Answer]: <list of objects>". Be comprehensive and avoid repeating my question. Use the given list of surrounding objects, maintaining the provided names. Only consider the surrounding objects in the given list. The objective is to identify all objects that could potentially block open actions. If an object obstructs the door or drawer from opening, include it in the final list of objects. Analyze the action movement and identify the blocking objects.

**Assistant:** Got it. I will output the reasoning process step-by-step, explain why I choose the object but not others and follow the output format.

**User:** [Query Object] + [Query Images]

**Assistant:** [Reply from GPT-4V]

Figure 7: **Prompts of the Decision-Making module.** We present the full prompts for the two pivotal roles of our decision-making module, **proposer** in (a), **verifier** in (b). The prompts are used for all our experiments without modification and extra examples.

### B.1.3 Decision-Making Module

As illustrated in the main paper, the decision-making module fulfills two crucial functions within our system. The first function serves as an action proposer (Fig. 7a), proposing the appropriate skill for the query object node. The subsequent role functions as the action verifier (Fig. 7b), tasked with confirming the feasibility of implementing the action and determining the action preconditions. The complete prompts for both roles are detailed in Fig. 7.

We adhere to the standard practice for designing prompts, as other papers do using LLM/LMM [18, 83, 89]. In order not to compromise the generalization ability of our system, we consistently use the same prompts across all scenarios and experiments. Our fundamental rule for prompt design is to minimize ambiguity and ensure alignment with our task. In our experiments, the average response time from GPT-4V is about 8 seconds for each question, which is acceptable as GPT-4V is only used in high-level task planning. For low-level motion planning, the use of action primitives allows us to meet the high-frequency requirement without having to constantly querying GPT-4V.

Our ACSG utilizes GPT-4V on every object node to progressively expand the graph. Hence, regardless of how complicated the scene is, each query posed to GPT-4V resides on a local node within our ACSG, essentially addressing the question, "Should I proceed with exploring this object, and if so, how?" As shown in our Matryoshka scenario, RoboEXP performs well in complex scenarios featuring five levels of hierarchical scene graphs and complicated exploration procedures. The commonsense knowledge learned by GPT-4V enables our system to efficiently explore the environment without having to manually design the exploration rules for diverse objects.

### B.1.4 Action Module

The action module focuses on providing useful action primitives to aid in constructing our ACSG. We have designed seven action primitives: "open the [door]", "open the [drawer]", "close the [door]", "close the [drawer]", "pick [object] to idle space", "pick back [object]", "move wrist camera to [position]". To fully support autonomous actions, we employ a heuristic-based algorithm leveraging geometric cues. The input to each action primitive is an object node of our ACSG, from which we can extract all necessary semantic, state, and low-level geometry information of the object. The integrated information can help us determine the specific grasping pose and path planning for opening and picking actions, which generalize to different instances in various positions and poses.

For opening action primitives related to doors or drawers, engagement with handles is required. In our implementation, we exploit the handle's position and geometry to discern its motion type (prismatic or revolute) and motion parameters (motion axis and motion origin). Executing this action involves utilizing the detected handle and its geometry to adeptly open doors or drawers. Upon identifying the specific handle to be operated, our system retrieves the point cloud converted from our voxel-based representation corresponding to that handle from our memory module. Subsequently, we employ Principal Component Analysis (PCA) to determine the principal direction of the handle, aiding in aligning the gripper for optimal engagement. Additionally, understanding the opening direction is pivotal for effectively handling doors or drawers. To ascertain this, we analyze neighboring points and deduce the most common normal as the opening direction. The combined information of the handle direction and the opening direction provides sufficient guidance for our robot arm to grasp the handle and open the prismatic part. However, in the case of a revolute joint, the motion becomes more intricate. Therefore, we further utilize the motion parameters inferred from the geometry to simulate the evolving opening direction based on the revolute joint's opening process. This well-designed heuristic empowers our system to reliably open drawers or doors in our tabletop setting.

For the pickup-related primitives, we simplify the pickup logic to exclusively consider a top-down direction. Consequently, our focus narrows down to acquiring essential information such as the object's height and xy location. We achieve this by extracting the object's point cloud from its associated voxel-based representation. Subsequently, we pinpoint the highest points within the cloud, calculating their mean to determine the optimal pickup point. This calculated point serves as a precise reference for our gripping mechanism, facilitating the successful grasping of objects in the specified direction.

Regarding viewpoint change, the primitive is parameterized with the expected pose. For example, after opening the door/drawer, to see inside, we develop the heuristic to choose the proper viewpoint from the open direction as the parameter for the primitive, allowing for the implementation of the action primitive.

### B.2 Other Design in Interactive Exploration

One desiderata for robot exploration is the ability to handle scenarios that necessitate multi-step or recursive reasoning. An example of this is the Matryoshka doll case, which cannot be addressed using previous one-step LLM-based code generation approaches [18, 19]. In contrast, our modular design allows agents to dynamically plan and adapt in a closed-loop manner, enabling continuous LLM-based exploration based on environmental feedback.

To manage multi-step reasoning, our system incorporates an action stack as a simple but effective "planning" module. Guided by decisions from the decision module, the stack structure adeptly organizes the order of actions. For instance, upon picking up the top Matryoshka doll, if the perception and memory modules identify another smaller Matryoshka doll in the environment, the decision module determines to pick it up. Our action stack dynamically adds this pickup action to the top of the stack, prioritizing the new action over picking back the previous, larger Matryoshka doll. This stack structure facilitates multi-step reasoning and constructs the system's logic in a deep and coherent structure.

Moreover, for the interactive scene exploration task, maintaining scene consistency is crucial in practice (e.g., the agent should close the fridge after exploring it). We employ a greedy strategy returning objects to their original states. This approach keeps the environment close to its pre-exploration state, making RoboEXP more practical for real-world applications.

### B.3  Usage of ACSG

The ACSG constructed during the exploration stage shows beneficial for scenarios that require a comprehensive understanding of scene content and structure, such as household environments like kitchens and living rooms, office environments, etc. We list several examples illustrating the potential usage of the scene graph in various tasks.

**Judging Object Existence.**  A direct application of our ACSG is to determine the presence or absence of specific objects in the current environment. For instance, during the exploitation stage of the scenario (App. D) to set the dining table, if the spoon is missing, the robot can further seek human assistance.

**Object Retrieval.** One notable advantage of our ACSG is its ability to capture all actions and their preconditions. Utilizing this information, retrieving any object becomes straightforward by following the graph structure and executing actions in topological order along the paths from the root to the target object node. For example, in the obstruction scenario (App. D), the ACSG can provide the sequence of actions required to fetch the tape: 1) removing the condiment blocking the cabinet door, 2) opening the cabinet via the door handle, and 3) retrieving the tape. Such insights are crucial for tasks like cooking.

**Advanced Usage.** The high-level representation of the environment provided by our ACSG serves as a simplified yet effective model. Similar to the approach proposed by Gu et al. [96], integrating the scene graph with Large Language Models (LLM) or Large Multi-modal Models (LMM) offers enhanced capabilities, including natural language interaction. This enables the robot to respond to human preferences expressed in natural language (e.g., fetching a coke when the person is thirsty) or through visual cues (e.g., fetching a mug when the table is dirty).

## C  Additional Details of Experiments

We conduct experiments in different settings to validate the effectiveness of the model. We provide additional experiments and results, including those with different lighting conditions and backgrounds, using different LMMs, intervention experiments, and several more room-level scenarios.

### C.1  Experiment Settings

Our experimental setup encompasses a diverse range of objects, as illustrated in Fig. 8. To assess the effectiveness of our system, we devised five types of experiments for the main quantitative results, each encompassing 10 distinct settings. These settings vary in terms of object number, type, and layout, as illustrated in Fig. 13.

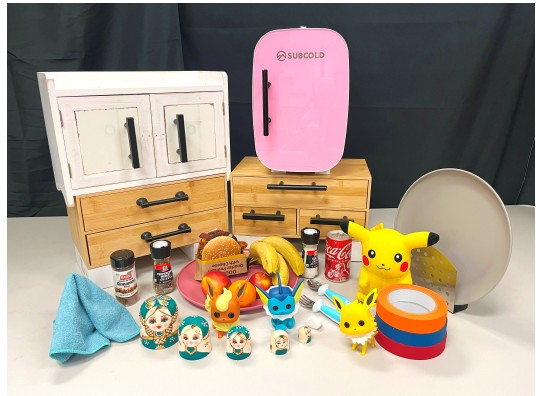

Figure 8: **All Testing Objects.** We present various objects utilized in our work, encompassing different types of cabinets, fruits, dolls, condiments, beverages, food items, tapes, tableware, and fabric.

### C.2  Evaluation

The details of five metrics employed for evaluation are as follows:

1) **Success:** This metric evaluates the success percentage across 10 variants for each task. We define success for each experiment as 1 when the final outputted ACSG exactly matches the GT version, and 0 otherwise.

2) **Object Recovery:** This metric quantifies the percentage of hidden objects successfully identified.

3) **State Recovery:** A binary value indicates whether the final state resembles the original state before exploration. This includes considerations for partial states and object positions (e.g., in the top drawer of a cabinet or on the table).

4) **Unexplored Space:** Evaluating the percentage of successfully explored need-to-explore space to reduce the robot's uncertainty about the scene. The identification of the need-to-explore space relies on human annotation.

5) **Graph Edit Distance (GED):** GED measures the disparity between the outputted graph and the GT graph. We adopt a simplified version of GED with six operations—three for nodes (add, delete, edit) and three for edges (add, delete, edit), with each operation incurring a cost of 1.

### C.3 Baselines

**GPT-4V Baseline:** We employ the pure GPT-4V as our baseline model along with the chain-of-thoughts (CoT) to enhance its capabilities, as outlined in a method similar to that proposed by Hu et al. [19]. The full prompt of the GPT-4V baseline is illustrated in Fig. 9.

**Heuristic Baseline:** We also design two heuristic baselines: i) *Heuristic-Open*: This method opens all handles; ii) *Heuristic-Full*: In addition to opening all handles, this method also picks up all movable objects to reveal what's underneath them.

**Random Baseline:** For the *Random* baseline, an action is chosen randomly for each object, including the option to take no action. Similar to the GPT-4V baseline, we make the heuristic and random baselines stronger by assuming that all actions are successful. We conducted the same set of experiments as in our original paper (5 task scenarios with 10 different settings for each scenario) for each of the baselines. All methods use the same perception module as our RoboEXP to understand the influence of different decision strategies.

**System:** You are an assistant tasked with aiding in the construction of a complete scene graph for a tabletop environment. The objective is to identify all objects hidden from the current observation in the tabletop setting. Your role involves selecting appropriate actions or opting not to take any action based on commonsense knowledge in response to queries with current observations. Your responses will guide a robot in efficiently exploring the environment. Approach each step thoughtfully, and analyze the fundamental problem deeply, considering the potential vagueness or inaccuracy in the queries. Adhere to the provided formats in your instructions.

**User:** Analyze and provide the current scene graph and your final answer for the next action given the latest observations in the tabletop scene from different viewpoints. Each time you need to pick an action to do or choose "Done" to terminate. The action you can choose should be composed of (<object/part>, <skill>). Be specific on which object or part you refer to. The skills you can choose: [1. Open the door. 2. Close the door. 3. Open the drawer. 4. Close the drawer. 5. Pick up the object to idle space. 6. Pick back the object from the idle space]. Each time after you choose an action, you will receive the new observations after the action. Format your responses as follows: "[Analysis]: <your reasoning process>; \n\n [Scene Graph]: <current scene graph> \n\n [Final Answer]: <skill>". Be comprehensive and avoid repeating my question. The primary goal is to select an action that has the potential to reveal hidden objects. The secondary goal is to act efficiently, performing only necessary actions to uncover hidden objects. The third goal is to make the object go back to the initial state after exploration. For the output scene graph, you need to output all the objects in the scene, including those found during the exploration process.

**Assistant:** Got it. I will output the reasoning process step-by-step, explain why I choose the skill but not others and follow the output format.

**User:** [Query Images]

**Assistant:** [Reply from GPT-4V]

**User:** [Query Images]

**Assistant:** [Reply from GPT-4V]

...

Figure 9: **Prompts of the GPT-4V baseline.** To ensure fairness in comparison to this baseline, we choose to use similar prompts, employing the chain-of-thoughts technique to enhance its performance.

#### C.3.1 Comparison to Heuristic & Random Baseline

We present all results in the above table in Tab. 2, which shows that our RoboEXP system is more capable and efficient in nearly all scenarios. The random baseline easily fails due to improper action selections. The *Heuristic-Open* baseline performs well in scenarios involving only open-action judgments but fails when pick actions are required. The *Heuristic-Full* baseline can provide better object recovery; however, its redundant pick actions create a scene graph with too many redundant

Table 2: **Quantitative Results on Different Tasks.** We compare the performance of baselines and our system across various tasks. Our system outperforms the baselines across nearly all metrics.

| Method | Success % ↑ | Object Recovery % ↑ | State Recovery % ↑ | Unexplored Space % ↓ | Graph Edit Dist. ↓ |
|---|---|---|---|---|---|
| **Drawer-Only** | | | | | |
| Random | 0±0.0 | 20±13.3 | 10±10 | 85±7.6 | 6.6±0.73 |
| Heuristic-Open | **90**±10.0 | **97**±3.3 | **100**±0.0 | **0**±0.0 | **0.2**±0.20 |
| Heuristic-Full | 0±0.0 | **97**±3.3 | **100**±0.0 | **0**±0.0 | 3.8±0.47 |
| GPT-4V | 20±13.3 | 83±11.0 | 60±16.3 | 15±7.6 | 2.8±1.04 |
| Ours | **90**±10.0 | **97**±3.3 | **100**±0.0 | **0**±0.0 | **0.2**±0.20 |
| **Door-Only** | | | | | |
| Random | 0±0.0 | 30±15.3 | 0±0.0 | 85±7.64 | 6.3±1.01 |
| Heuristic-Open | **90**±10.0 | **100**±0.0 | **100**±0.0 | **0**±0.0 | **0.1**±0.10 |
| Heuristic-Full | 0±0.0 | **100**±0.0 | **100**±0.0 | **0**±0.0 | 3.8±0.47 |
| GPT-4V | 30±15.2 | 50±16.7 | 80±13.3 | 40±14.5 | 4.4±1.52 |
| Ours | **90**±10.0 | **100**±0.0 | **100**±0.0 | **0**±0.0 | **0.1**±0.10 |
| **Drawer-Door** | | | | | |
| Random | 0±0.0 | 4±4.0 | 0±0.0 | 85±6.7 | 12.8±0.80 |
| Heuristic-Open | **70**±15.3 | **91**±4.7 | **100**±0.0 | **0**±0.0 | **0.5**±0.27 |
| Heuristic-Full | 0±0.0 | **91**±4.7 | **100**±0.0 | **0**±0.0 | 6.1±0.71 |
| GPT-4V | 10±10.0 | 62±10.7 | 70±15.3 | 25±6.6 | 5.6±1.46 |
| Ours | **70**±15.3 | **91**±4.7 | **100**±0.0 | **0**±0.0 | **0.5**±0.27 |
| **Recursive** | | | | | |
| Random | 10±10.0 | 33±12.9 | 60±16.3 | 70±12.8 | 8.8±1.91 |
| Heuristic-Open | 0±0.0 | 0±0.0 | **100**±0.0 | 100±0.0 | 10±1.69 |
| Heuristic-Full | 40±16.3 | **90**±10.0 | **100**±0.0 | **5**±5.0 | **1.8**±0.63 |
| GPT-4V | 0±0.0 | 20±13.3 | 70±15.3 | 63±15.3 | 8.8±2.06 |
| Ours | **70**±15.3 | 80±11.7 | **100**±0.0 | 15±8.9 | 2.1±1.49 |
| **Occlusion** | | | | | |
| Random | 0±0.0 | 0±0.0 | 0±0.0 | 100±0.0 | 9.8±0.53 |
| Heuristic-Open | 0±0.0 | 22±11.7 | 0±0.0 | 75±8.3 | 8.6±0.72 |
| Heuristic-Full | 0±0.0 | 22±11.7 | 0±0.0 | 75±8.3 | 9.4±0.58 |
| GPT-4V | 0±0.0 | 17±11.4 | 10±10.0 | 85±7.6 | 7.3±0.97 |
| Ours | **50**±16.7 | **67**±14.9 | **70**±15.3 | **30**±15.3 | **2.5**±1.15 |

Table 3: We report the **average number of actions** for settings in which all methods find all objects, explore all hidden spaces, and recover the states before exploration. In the recursive and occlusion scenarios, some baseline methods do not have experiments that meet these requirements. Our RoboEXP consistently requires fewer actions compared to the baselines and is more aligned with the ground truth of human action choices.

| Method | Drawer-Only | Door-Only | Drawer-Door | Recursive | Occlusion |
|---|---|---|---|---|---|
| GT (Human) | 4.0 | 4.0 | 8.0 | 4.6 | 6.0 |
| Heuristic-Full | 7.0 | 8.0 | 12.0 | 6.0 | - |
| GPT-4V | 6.0 | 4.4 | 8.0 | - | - |
| Ours | **4.0** | **4.0** | **8.0** | **4.6** | **6.3** |

nodes, resulting in higher Graph Edit Distance in most scenarios. In complicated scenarios like object-object occlusion, the heuristic baselines completely fail, as they do not reason about object-object relationships as we do using the action verifier in our system.

There are two main reasons why heuristic-based baselines are not ideal: i) *Defining all potential heuristics to cover every scenario is nearly impossible.* Specifying and enumerating all semantic categories is tedious, and some size-based heuristics can also fail (e.g., a threshold that is too small may filter out objects that should be picked up, such as clothes on a Kindle, while a threshold that is too large may include non-movable objects); ii) *Baselines with manually defined heuristics can create excessive redundant actions.* They tend to pick up everything, making exploration inefficient. In fact, our RoboEXP can be viewed as a special form of a frontier-based method with LMM-guided action selection, which demonstrates better efficiency and effectiveness in our interactive exploration task.

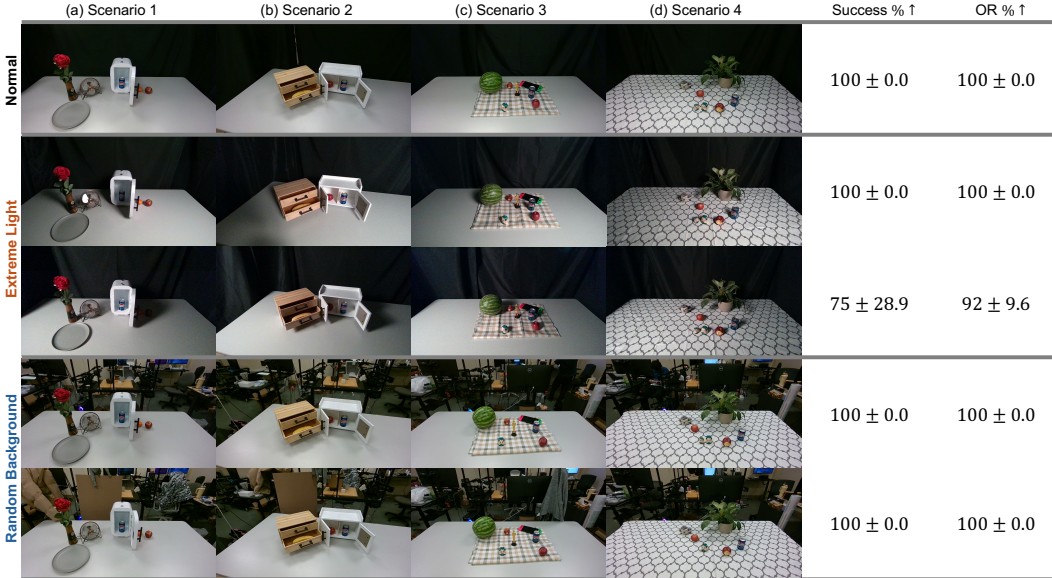

| | (a) Scenario 1 | (b) Scenario 2 | (c) Scenario 3 | (d) Scenario 4 | Success % ↑ | OR % ↑ |
|---|---|---|---|---|---|---|
| Normal | | | | | 100 ± 0.0 | 100 ± 0.0 |
| Extreme Light | | | | | 100 ± 0.0 | 100 ± 0.0 |
| | | | | | 75 ± 28.9 | 92 ± 9.6 |
| Random Background | | | | | 100 ± 0.0 | 100 ± 0.0 |
| | | | | | 100 ± 0.0 | 100 ± 0.0 |

Figure 10: **Experiments on Extreme Illumination and Random Background.** We conduct experiments in four scenarios with varying lighting conditions and random backgrounds. The reported numbers are averages over four scenarios for each condition. Our system performs well across all conditions. (OR refers to our Object Recovery metric).

Table 4: **Quantitative Results on Different LMMs.** We conduct experiments with GPT-4V and LLaVA acting as the core of the RoboEXP decision module, under the same fifteen settings as in Fig. 10.

| Metric | Success % ↑ | Object Recovery % ↑ | State Recovery % ↑ | Unexplored Space % ↓ | Graph Edit Dist. ↓ |
|---|---|---|---|---|---|
| Ours (LLaVA) | 25±25.6 | 50±29.6 | **100**±0.0 | 23±21.9 | 2.5±0.98 |
| Ours (GPT-4V) | **95**±12.9 | **98**±4.3 | **100**±0.0 | **0**±0.0 | **0.1**±0.26 |

## C.3.2 Comparison of Efficiency

The number of actions is only comparable when all hidden objects are found, all unexplored spaces are fully explored, and all object states are recovered after exploration. Given that the *Random* and *Heuristic-Open* baselines do not provide comparable exploration results, we primarily compare the efficiency of *Heuristic-Full*, GPT-4V, and our RoboEXP system. We report the average number of actions in settings where all methods find all hidden objects, fully explore the unexplored space, and restore the object states after exploration in the above table in Tab. 3. The results show that our RoboEXP consistently takes fewer actions to construct the full ACSG during the interactive exploration process and is more aligned with human action choices.

## C.4 Extreme Illumination and Random Background

RoboEXP is robust to extreme lighting conditions and complex backgrounds. To demonstrate this, we tested under four different scenarios, each with varying lighting conditions and random backgrounds. Fig. 10 shows twenty different settings and their corresponding results. In various conditions and scenarios, our system is able to successfully conduct interactive exploration and construct the ACSG, indicating the robustness of RoboEXP to these factors.

## C.5 Performance on Different LMMs

RoboEXP is compatible with different multimodal foundation models beyond GPT-4V. We conducted additional experiments using the latest LLaVA-v1.6-34b as the core of our decision module and compared it against GPT-4V under the same settings. Tab. 4 shows that both models can work with

our RoboEXP system, yet the capacity of LMMs does influence the overall performance. In general, GPT-4V achieves a higher success rate and more consistent behaviors across different scenarios.

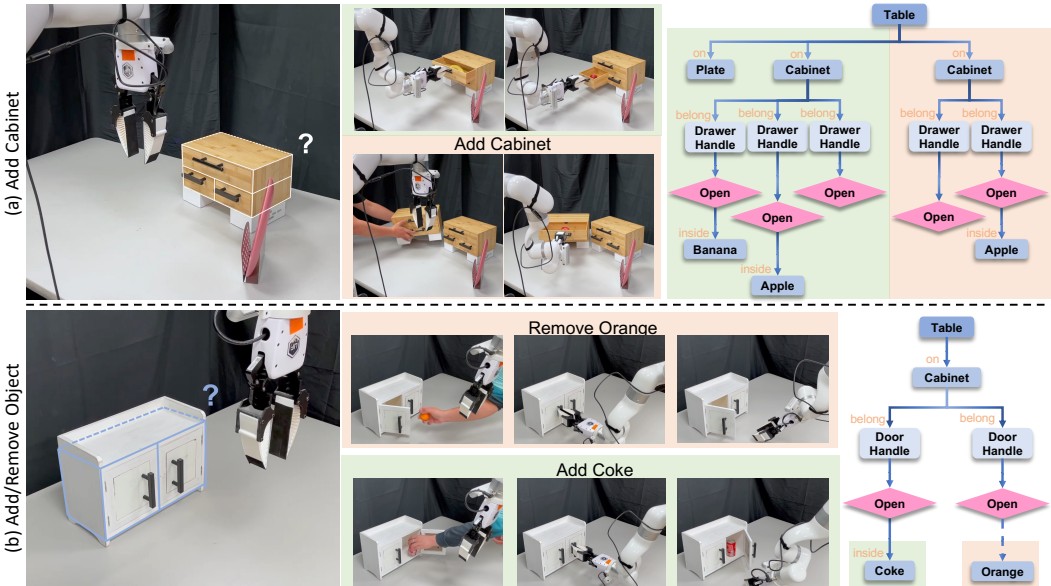

Figure 11: **Qualitative Results on Different Intervention Scenarios.** (a) This scenario involves adding a cabinet to the tabletop setting, and our system can auto-detect the new cabinet and explore the objects inside. (b) This scenario includes removing and adding objects from and into the cabinet. Our system can monitor hand interactions and re-explore the corresponding doors.

## C.6    Human Intervention

Our RoboEXP system possesses the capability to autonomously adapt to changes in the environment. We employ two types of human interventions to demonstrate these points (refer to App. D).

The first type of intervention (Fig. 11a) involves adding new cabinets to the scene. In this scenario, we add a cabinet to the explored area, allowing our system to automatically explore the newly added cabinets and update the ACSG.

The second type of intervention (Fig. 11b) involves adding new objects to or removing existing ones from the cabinets in the current scene. Our system can monitor human interactions and discern which objects require re-exploration. Subsequently, it autonomously updates the ACSG based on re-exploration.

## C.7    Room-Level Household Scenarios

RoboEXP can work well in room-level household environments. To demonstrate this, we conducted two experiments within an apartment (see Fig. 12), specifically in the dining area and bedroom. We integrated four RGB-D observations captured by a handheld RealSense D455 with ICP-based multi-way alignment. Our system successfully constructs corresponding scene graphs within the room-level household environments. Once the static scene graph is constructed, our decision module effectively identifies the correct objects for exploration. Specifically, it accurately identifies the fridge in Fig. 12 (a) and the cabinet in Fig. 12 (b) for further exploration. Tab. 5 shows the complete responses from GPT-4V in our decision module on determining the actions to take in our two household scenarios.

## D    Video Timeline

### Scenario A. Exploration-Exploitation

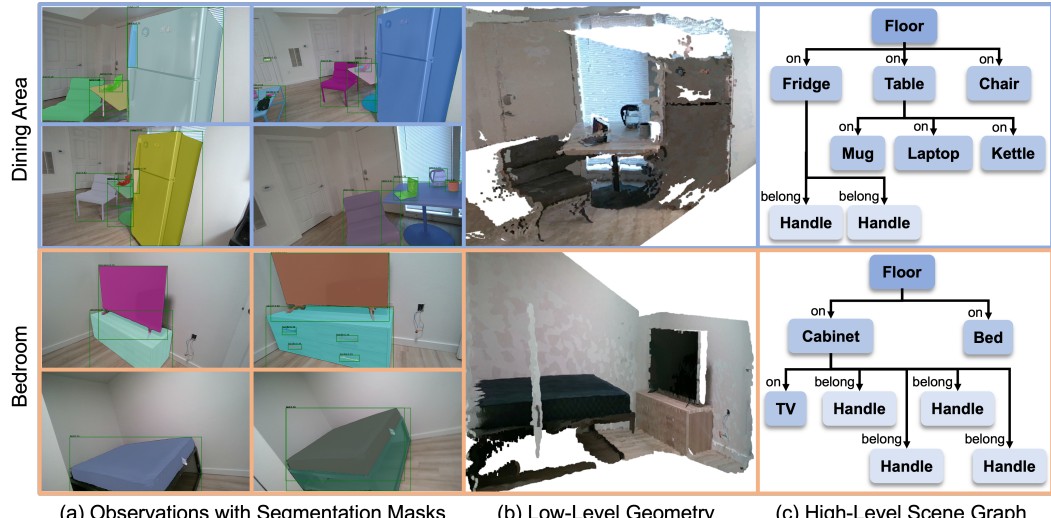

(a) Observations with Segmentation Masks     (b) Low-Level Geometry     (c) High-Level Scene Graph

Figure 12: **Experiments on Room-Level Household Scenarios.** We conduct experiments in two room-level household environments. The dining area includes a table, fridge, and items on the table, whereas the bedroom includes a bed, cabinet, and a TV. The figure presents (a) the observations with segmentation masks; (b) the low-level reconstructed geometry; (c) the built high-level scene graph.

Exploration: 00:43 - 01:16

Exploitation: 01:17 - 01:37

**Scenario B. Recursive Reasoning**

Exploration: 01:49 - 02:26 (Two scenarios)

**Scenario C. Obstruction**

Exploration: 02:33 - 02:59

**Scenario D. Intervention**

Exploration: 03:05 - 04:09 (Two scenarios)

**Extension to Mobile Robot**

Exploration: 04:13 - 04:53

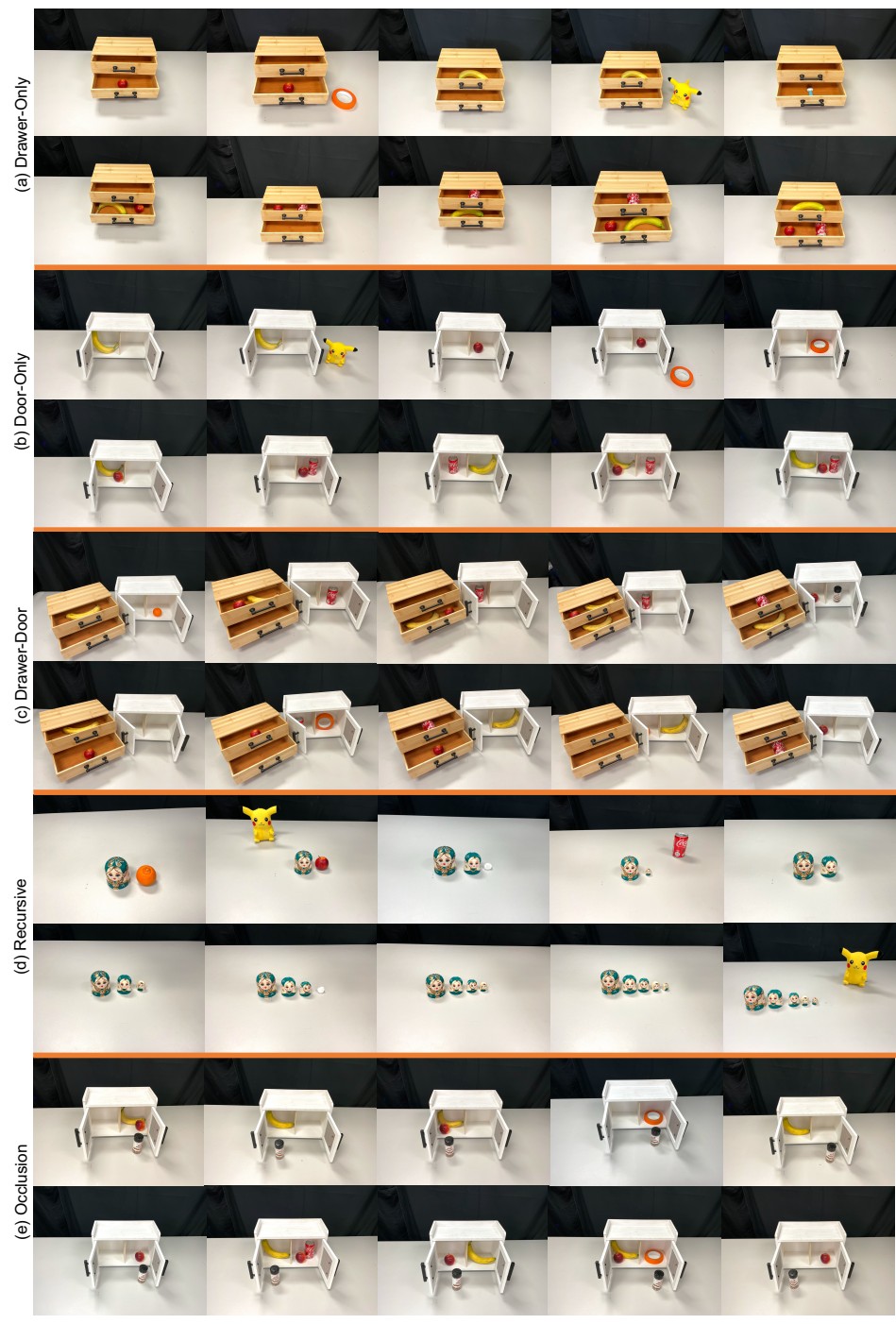

Figure 13: **Experiment Settings.** Varied object numbers, types, and layouts in our experimental settings of the quantitative results.

Table 5: **Decision Module Logs for Household Scenarios.** We present all responses from GPT-4V for our two household scenarios: the dining space case and the bedroom case. The inputs to our decision module are images from various viewpoints of the query object in a green bounding box.

| Query Object | LMM Response |
|---|---|
| **Dining Space** | |
| Fridge | **[Analysis]:** The object in question is a fridge, which is typically used to store food and drinks. The green bounding box in the images highlights the fridge, which has doors that can be opened. Since the goal is to identify all objects hidden from the current observation in the tabletop setting, and a fridge is an appliance that commonly contains items inside, it is reasonable to assume that opening the fridge could reveal hidden objects. 
 **[Final Answer]:** Open the doors or drawers. |
| Chair | **[Analysis]:** The object in question is a chair, which is typically not used to store objects in a way that would require opening or removing parts to reveal them. Chairs are designed for sitting, and while it is possible for items to be placed on or under them, the chair itself does not have compartments that would be opened or closed. The images show the chair from different angles, and there is no indication that there are hidden compartments or objects on the chair that would necessitate an action to uncover them. The chair does not have a top object that would be opened or picked up, and it does not have doors or drawers. 
 **[Final Answer]:** No action. |
| Table | **[Analysis]:** The object in question is a table, which is highlighted with a green bounding box in the images provided. From the different viewpoints, it is clear that the table has a flat top surface with no visible drawers or doors that could be opened to reveal hidden objects. The table does not appear to have a covering that could hide something underneath it. The table's design is simple, with a single central support and no storage compartments. Therefore, there is no action required that would potentially reveal hidden objects in this case. 
 **[Final Answer]:** No action. |
| Laptop | **[Analysis]:** The object in question is a laptop, which is placed on a table. Laptops do not have a hidden compartment that would typically contain other objects, nor do they obstruct the view of objects underneath them when they are in an open position, as seen in the images. The laptop is already open, and there is no indication that there is anything on top of it or that it is covering anything beneath it. Therefore, there is no action required that would reveal any hidden objects in this case. 
 **[Final Answer]:** No action. |
| Mug | **[Analysis]:** The mug in the green bounding box is sitting on a tabletop and does not appear to be covering anything, nor does it have a lid that could be hiding something inside. Mugs are typically open containers and do not have compartments that would conceal objects. Therefore, there is no need to pick it up to look underneath. 
 **[Final Answer]:** No action. |
| Kettle | **[Analysis]:** The object in question is a kettle, which is sitting on top of a table. Kettles are typically not used to store other items, especially when they are placed in a setting like a tabletop where they are likely used for their intended purpose of boiling water. The kettle does not have drawers or doors, and it is unlikely that it is obscuring any items on the table due to its shape and size. Therefore, there is no need to open any part of the kettle, and it is not obstructing any items underneath it. 
 **[Final Answer]:** No action. |
| **Bedroom** | |
| Cabinet | **[Analysis]:** The object in question is a cabinet, as indicated by the green bounding box in the provided images. Cabinets typically have doors or drawers that can be opened to reveal contents stored inside. In the images, the cabinet appears to have several drawers. Since the primary goal is to select an action that has the potential to reveal hidden objects, and the cabinet has drawers that could contain items, the appropriate action would be to open these drawers. 
 **[Final Answer]:** Open the doors or drawers. |
| Bed | **[Analysis]:** The object in question is a bed, which is typically not a storage furniture piece with doors or drawers. However, some beds do come with storage options underneath, such as drawers or a lifting mechanism for accessing a storage space. In the provided image, the bed appears to have a solid base without visible drawers or doors. There is no indication that the bed has a mechanism to lift the mattress to reveal storage underneath. Therefore, it is unlikely that there are hidden objects within the bed that can be accessed by opening doors or drawers. 
 **[Final Answer]:** No action. |
| Television | **[Analysis]:** The object in question is a television, which is placed on top of a surface, likely a media console or a dresser with drawers. Televisions are not typically used to store objects behind or within them, as they are electronic devices designed for display purposes. The images show the television from two different angles, and there is no indication that there are any hidden compartments or objects behind the screen that would be revealed by moving it. Additionally, televisions are generally not designed to have openable parts that would conceal objects. The media console or dresser beneath the television does have drawers, but since the query is specifically about the television, these are not the focus for the action to be taken. 
 **[Final Answer]:** No action. |

