# OpenReview forum: "RoboEXP: Action-Conditioned Scene Graph via Interactive Exploration for Robotic Manipulation"
_robot-learning.org/CoRL/2024/Conference — CoRL 2024_

### Official Review · Reviewer_5sxQ · 2024-07-20
**Action-conditioned scene graph by exploration**

**Originality:** 3
**Technical Quality:** 4
**Clarity Of Presentation:** 4
**Potential Impact:** 3
**Recommendation:** 3
**Confidence:** 5

**Review:**

Overall, the idea of interactive exploration is compelling and well-presented. The methods are clear and the experiments contain multiple experiments on two real robot systems. Some points of the methodology could be more clearly stated. The work would benefit from a clearer throughline, backed up with experiments, regarding the nature of the contribution. Detailed comments are below.

**Strengths:**
* The experimental analysis is thorough, providing multiple metrics to understand different elements of the method. The error breakdown is very helpful in understanding the success rate of the method and is a valuable inclusion.
* The method shows success on two real robot platforms in exploration and various task execution. The experiments involve interacting with multiple objects including articulated and deformable ones.
* The paper is presented and explained clearly with nice visuals and data presentation.

**Points of feedback:**

It is a bit unclear whether the contribution is the scene representation or the exploration framework. The experiments focus on the exploration framework, but are lacking in baselines or ablations to demonstrate the key benefits of the RoboEXP framework. For example, does the proposed decision module outperform a brute force approach (perform every possible detected action)? Such a heuristic baseline is akin to the classic frontier exploration strategy. Another ablation could be a random approach (randomly sample viewpoints or actions). I suspect that a frontier exploration-style approach would perform very well on most metrics.

A stated goal of the work is to explore with minimum time (Section 3, line 112). While a strength of the paper is the inclusion of multiple metrics in the evaluations, it would be beneficial to show the time or number of actions taken to create the scene graph to support this claim.

Finally, some fundamental assumptions from the paper are missing in order to make a compelling case for the generalization capability of the proposed representation. First, it would be helpful to elaborate on the taxonomy of relations and actions and how they were chosen, as this impacts to which tasks and scenes this method can be applied. Second, it seems the representation proposed would be dependent on the state (e.g. in Fig 5, when the drawer is already open, the “condiment” can be picked without the “open” action. Is the graph dynamically updated in this case? Third, an action is not a property of an object alone but instead depends on both the agent and object (e.g. a robot may not be able to grasp an object that is too large or too heavy). How are these cases handled by the proposed method?

*Other points of feedback:*
* What is the definition of Graph Edit Distance? How is it computed?
* The baseline would benefit from a bit more detail. Is the robot teleoperated for the baseline?
* What is the importance of the state recovery metric? This might not always be important to the task.
* Is the given ACSG unique? There may be more than one possible action to reveal an occluded object (e.g. change viewpoint or move an object).
* Is it always necessary to construct the whole scene graph (e.g. when objects unrelated to the task at hand are present in the scene)? This may be an interesting line of future work if not already handled. The authors may consider related works in graph-based object search [1] and semantic search [2].

**References:**

[1] Zhen Zeng, Adrian Röfer, Odest Chadwicke Jenkins. SLiM: Semantic Linking Maps for Active Visual Object Search. ICRA 2020.

[2] Naoki Yokoyama, Sehoon Ha, Dhruv Batra, Jiuguang Wang, Bernadette Bucher. VLFM: Vision-Language Frontier Maps for Zero-Shot Semantic Navigation. ICRA 2024.

**Quality Of The Limitations Section:**

3

**Questions For Rebuttal:**

1. How does RoboEXP compare to other exploration heuristic baselines, like random or frontier exploration?
2. How much time (or how many actions, as is relevant) is needed to construct the scene graph?
3. What are the assumptions about the environment and type of actions?

See additional details and points above.

**Robotics Focus:**

4

**Summary Of Paper:**

This paper presents a scene representation called the Action-Conditioned Scene Graph, which represents both inter-object spatial relationships and the necessary actions to reveal elements of the scene graph. The paper then describes an exploration framework which constructs the proposed perceptual representation through interaction. Experiments on real-world robotic manipulation tasks demonstrate increased success rate and improved exploration metrics compared to a baseline heuristic approach.

**Summary Of Recommendation:**

The paper is clearly presented and has been evaluated on real robots. The concept of the action-conditioned scene graph is well motivated. However, more clarity is needed regarding the assumptions of the system in order to generalize across scenarios. Furthermore, comparison against heuristic baselines would strengthen the claims of the work.

---

### Official Review · Reviewer_hwZM · 2024-07-20

**Originality:** 4
**Technical Quality:** 3
**Clarity Of Presentation:** 3
**Potential Impact:** 3
**Recommendation:** 3
**Confidence:** 3

**Review:**

This paper presents an approach to reason about interactions through an action-conditioned graph. The work is well written and clearly motivated.

Strengths:

The paper does a good work at illustrating the overall system. The results seem quite promising, with clearly higher success rates in various different metrics. The qualitative results also show impressive interaction and discovery in the environment.

Weaknessness:

There is a bit of clarity in the exact robotic implementation that should be addressed in the main document. In particular, how the robot goes about interacting and whether the actions are needed to be designed by the user a-priori for the approach to work. In addition, some of the research questions that the results address could be more specific e.g., what efficiency is expected in exploration from the action-conditioned graph, how quickly does the approach explore all interactions, etc. Furthermore, the paper could be clearer about some metrics, specifically the similarity/differences, meanings, and implications that the metrics hold.

**Quality Of The Limitations Section:**

3

**Questions For Rebuttal:**

Does the user need to specify all the interactions a-priori? It seems that one can interact with an object an infinite number of ways, so how does the graph, which is finite, deal with this?

What does it mean to take an action that reduces the unexplored node set? Specifically, in a real-world application, how does an action yield a node from the environment. Is this part of the perception module?

How does the agent know when the exploration is terminated? If it has access to the unexplored node-	set, then why explore it at all and enumerate each node?

What is the ACSG “output”?

The DEG metric and unexplored space metric seem like they would produce similar meanings. Could their differences and implications ve explained more clearly?

**Robotics Focus:**

4

**Summary Of Paper:**

The paper presents a novel way to represent interactions in the environment using action-conditioned scene graphs and exploration for manipulation problems.

**Summary Of Recommendation:**

The work seems promising, but requires some clarifications within the text.

---

### Official Review · Reviewer_Pzsb · 2024-07-22
**review for submission 192**

**Originality:** 2
**Technical Quality:** 2
**Clarity Of Presentation:** 4
**Potential Impact:** 2
**Recommendation:** 3
**Confidence:** 4

**Review:**

In this paper, the authors propose a RoboEXP system, which generates an action-conditioned scene graph via interactive exploration and the LMM is incorporated. The generated scene graph is expected to facilitate a bunch of downstream manipulation tasks. The structure of the paper is clear, and it is good to the real world experiments. Here are some concerns.

- The idea of interacting with the environment to generate the scene graph is reasonable. However, the scenario presented in this paper is actually artificial. For example, we seldom put one fruit in one drawer, and only one coke in the refrigerator in reality. It is said that the LMM could bring commonsense knowledge for decisions. I wonder how much the commonsense knowledge could be leveraged in such situations.

- In the experiments, the proposed system and the GPT-4V methods are compared and an obvious improvement can be observed. It is suggested that a comparison of efficiency of each method is also presented or discussed. Additionally, some heuristic methods such as opening every door in the view to see what is in it could be included.

- For the downstream manipulation tasks, does the robot uses a pre-generated scene graph or with an online scene graph generation process. The analysis about the application of the scene graph is not sufficient.

**Quality Of The Limitations Section:**

2

**Questions For Rebuttal:**

- The setup of the scenario and how the setup and commonsense knowledge affect the results.

- The efficiency of the comparison methods. And some heuristic method is suggested.

- The process of scene graph generation in the task.

**Robotics Focus:**

4

**Summary Of Paper:**

In this paper, the authors propose a RoboEXP system, which generates an action-conditioned scene graph via interactive exploration and the LMM is incorporated.

**Summary Of Recommendation:**

The generalization of the task to real-word environment and some details of the method are not quite clear.

---

### Author Rebuttal · Authors · 2024-08-07

Our revised main paper and supplement are in the rebuttal file.

---

### Decision · Program_Chairs · 2024-09-04

**Decision:**

Accept

**Comment:**

This paper proposes an approach to generate action-conditioned scene graphs and uses it for interactive exploration for several manipulation tasks. The reviewers find the paper well written, find the results promising, and appreciate the real-world experiments. However, the reviewers have also raised several concerns including lacking baselines or ablations, unrealistic scenarios in the experiments, insufficient analysis on application of the scene graph, lacking details on the proposed method, require clarity on the research questions, clarity on the metrics used,  lacking clarity regarding the assumptions of the system, among others. Each of the reviewers has listed several questions to be addressed in the rebuttal.

Post-rebuttal: Most of the concerns of the reviewers have been sufficiently addressed in the rebuttal. The paper could be stronger if results were presented in more complex realistic scenes.